# Temperature Effects on Sulfuric Acid Aerosol Nucleation and Growth: Initial Results from the TANGENT Study

Lee Tiszenkel[1], Chris Stangl[2], Justin Krasnomowitz[2], Qi Ouyang[1], Huan Yu[3], Michael J. Apsokardu[2], Murray V. Johnston[2], Shan-Hu Lee[1,4,*]

[1]Department of Atmospheric Science, University of Alabama in Huntsville, Huntsville, AL
[2]Department of Chemistry and Biochemistry, University of Delaware, Newark, DE
[3]School of Environmental Science and Engineering, Nanjing University of Information Science and Technology, Nanjing, China
[4]Department of Environmental Science and Engineering, Fudan University, Shanghai, China

*Correspondence to*: Shan-Hu Lee (shanhu.lee@uah.edu)

**Abstract.** New particle formation (NPF) consists of two steps: nucleation and subsequent growth. At present, chemical and physical mechanisms that govern these two processes are not well understood. Here, we report initial results obtained from the TANGENT (Tandem Aerosol Nucleation and Growth Environment Tube) experiments. The TANGENT apparatus enables us to study these two processes independently. The present study focuses on the effects of temperature on sulfuric acid nucleation and further growth. Our results show that lower temperatures enhance both the nucleation and growth rate. However, under temperatures below 268 K the effects of temperature on the nucleation rate become less significant and the nucleation rate becomes less dependent on RH, indicating that particle formation takes place via barrierless nucleation at lower temperatures. We also examined the growth of newly formed particles under differing temperature conditions for nucleation and further growth. Our results show that newly nucleated clusters formed at low temperatures can indeed survive evaporation and grow in a warmer environment in the presence of $SO_2$ and ozone. These results also imply that some heterogeneous reactions involving nanoparticles affect nucleation and growth of newly formed particles.

## 1. Introduction

Atmospheric nanoparticles affect human health and air quality. Newly formed particles can contribute to approximately 30-70% of cloud condensation nuclei (CCN) in the atmosphere (Merikanto et al., 2009;Wang and Penner, 2009;Yu and Luo, 2009;Gordon et al., 2017). NPF takes place via two steps: initial nucleation (formation of critical clusters) and subsequent growth of nucleated clusters (Kulmala et al., 2013). At present, chemical and physical mechanisms that govern these two processes, as well as the identity of chemical precursors involved in these processes, are still not well understood (Zhang et al., 2012;Yu et al., 2017b). Current global models fail to represent NPF in the atmosphere for a wide range of temperature and RH conditions and for different emissions of biogenic and anthropogenic precursors due to a lack of observations. For example, models predict frequent NPF during the summer in mixed deciduous forests in the United States (Yu et al., 2015), while field

observations show an absence of NPF in this region (Kanawade et al., 2011;Lee et al., 2016). Also, current NPF theories are unable to explain the frequent NPF observed in extremely polluted megacities (Yu et al., 2017b;Kulmala et al., 2017).

Temperature and RH are the key thermodynamic properties of aerosol formation and growth (Seinfeld and Pandis, 2016). Nucleation rate ($J$) is a function of temperature and the Gibbs free energy barrier of cluster formation. At lower temperatures, Gibbs free energy barriers become lower and critical cluster diameters become smaller. Condensational species can affect aerosol growth differently at different temperatures because their saturation vapor pressures are dependent on temperature. For water, RH is the same as saturation ratio and chemical activity. Laboratory experiments of aerosol nucleation and growth as a function of temperature and RH remain limited, although these observations are critically needed in global models to correctly parameterize NPF under various altitude, latitude and seasonal conditions. Aerosol nucleation experiments are extremely challenging due to various experimental difficulties, including contamination of base compounds (Erupe et al., 2011;Yu et al., 2012). At present there is a lack of consistency between different experiments from different groups and even from the same groups using the same experimental setup. The lack of reproducibility and consistency of the nucleation experiments greatly hinders our understanding of nucleation mechanisms.

(Duplissy et al., 2016) conducted studies of binary homogeneous nucleation of sulfuric acid and water, with and without ions in the CLOUD (Cosmics Leaving OUtdoor Droplets) chamber at different temperatures ranging from 207 to 299 K and RH between 11% - 58%. At lower temperatures both ion nucleation and neutral binary nucleation are at the kinetic regime, while at higher temperatures $J$ is strongly dependent on [$H_2SO_4$], indicating there are high Gibbs free-energy barriers at these temperatures. At the nucleation regime, nucleation rates are strongly dependent on RH. Kürten et al. (2016) reported the temperature dependence of ternary nucleation in the CLOUD chamber, at the temperature from 208 to 298 K, [$H_2SO_4$] between $10^5$ and $10^9$ cm$^{-3}$, and [$NH_3$] up to ~1400 pptv. At 208 K, $J$ reached the threshold of 1 cm$^{-3}$ s$^{-1}$ at a [$H_2SO_4$] of ~$3\times10^6$ cm$^{-3}$ for the binary case, and at [$H_2SO_4$] of ~$510^5$ cm$^{-3}$ for the ternary case with [$NH_3$] of 5 pptv. At 298 K, RH has strong effects on the measured $J$ for both charged and neutral ternary nucleation, because the increase in RH could lead to a displacement of $NH_3$ from the stainless-steel walls in the CLOUD chamber and lead to an elevated $NH_3$ background level and consequently to higher $J$.

Laboratory experiments of growth rates ($GR$) of newly nucleated particles are very sparse. (Skrabalova et al., 2014) studied $GR$ of newly formed particles in a flow tube at the temperature between 283 to 303 K and RH of 1% and 30%, designated as "dry" and "wet" conditions, respectively. [$H_2SO_4$] was varied between $2\times10^8$ cm$^{-3}$ and $1.4\times10^{10}$ cm$^{-3}$. They found different effects of RH on $GR$; at [$H_2SO_4$] below $10^9$ cm$^{-3}$, growth is promoted in drier conditions, whereas at [$H_2SO_4$] higher than $10^9$ cm$^{-3}$, growth favours wetter conditions. Yu et al. (2017a) performed flow tube experiments of sulfuric acid aerosol nucleation, at the temperature from 248 to 313 K and RH from 1% to 79%, under minimal base concentrations ([$NH_3$] < 23 pptv, methylamine < 1.5 pptv, and dimethylamine < 0.52 pptv). This study provides for the first time the temperature and RH dependence of both $J$ and $GR$. $J$ shows the following dependence within the experimental conditions:

$$J = 10^{41.8}[RA]^3[RH]e^{\frac{-2.4\times10^4}{T}}, \tag{1}$$

where RA is relative acidity (or saturation ratio) of sulfuric acid, and T temperature. Their results show that *GR* is independent of temperature below 290 K, but significantly decreases at temperatures above 290 K. RH has a moderate effect on *GR*.

At present, there is still no clear evidence from atmospheric observations that demonstrates "causal effects" of temperature and RH on NPF. Recent CLOUD studies have shown that oxygenated organics formed from BVOCs can grow newly nucleated particles in a wide range of tropospheric temperatures (Stolzenburg et al., 2018). This is because gas phase autooxidation reactions involved in the formation of HOMs usually have strong temperature dependencies, with higher reaction rates at higher temperatures (Frege et al., 2018), whereas nucleation is favoured at lower temperatures.

Another important perspective of temperature effects on NPF is the effects of temperature on evaporation of newly nucleated clusters and nanoparticles that have undergone transport to differing temperature conditions for further growth. Boundary layer particle concentrations in the marine regions have been shown to be a result of sulfuric acid-driven nucleation of nanoparticles in the free troposphere as a result of oxidation of upwardly transported and oceanic dimethylsulfide. These particles then undergo downward transport to become reservoirs of nanoparticles in the boundary layer. This has been observed and modelled by several groups over the previous decades (Russell et al., 1994;Raes, 1995;Clarke, 1993).

The lack of local NPF events in Amazon forests has been a confounding observation for many years (Rizzo et al., 2018) and yet there are reservoirs of nuclei mode particles at the surface which do not form typical "banana" plots of aerosol size distributions observed elsewhere (Martin et al., 2016). Understanding the origin of these nuclei mode particles has been a subject of recent studies. (Wang et al., 2016) reported that while NPF does not take place in Amazon forests at the surface level during the dry and wet season, NPF takes place in the colder free troposphere; these newly formed particles can be transported down to the boundary layer to become a reservoir of nanoparticles at the surface. This downward transport process is very similar to the above-mentioned marine boundary layer nanoparticle processes (Russell et al., 1994; Raes et al., 1995; Clarke et al., 1993). A subsequent question is thus whether newly nucleated clusters and nanoparticles in the free troposphere can survive evaporation during their transport to the warmer boundary region.

At present, the temperature effects on the growth of newly formed particles crossing different temperature regions have not been examined in a controlled laboratory environment. In order to address this shortcoming in laboratory experiments of this relatively common phenomenon, we have constructed the TANGENT apparatus, consisting of a temperature- and RH-controlled nucleation flow tube that permits study of gas-to-particle conversion over a wide parameter space. This nucleation region is connected to a room temperature growth tube where conditions for further growth of these nanoparticles can differ dramatically from those of their nucleation. With the suite of instruments monitoring precursor gases as well as particle size distributions from sub-3 nm up to CCN sizes at key areas of the flow apparatus, the critical processes that atmospheric nanoparticles undergo can be studied and parameterized with TANGENT.

Here, we present the initial results of the TANGENT experiments conducted during the Intensive Observation Period (IOP) study in June and July 2018. In the present study, we report the temperature effects on *J* and *GR* of sub-2 nm particles in the nucleation tube. We also discuss how temperature differences in the nucleation and growth tube affect the potential evaporation of newly nucleated clusters while these clusters are transported from the colder to the warmer temperature region.

## 2. Methods

### 2.1 The TANGENT Experimental Setup

Figure 1 shows the schematic diagram of the TANGENT apparatus. The TANGENT consists of the two flow tubes (FT) to enable studies of nucleation (FT-1) and subsequent growth (FT-2). The nucleation tube was built by the University of Alabama in Huntsville, and the growth tube by University of Delaware. TANGENT allows for nucleation to be observed as a separate process from growth; with a short residence time (45 s), precise temperature control, and the ability to produce $H_2SO_4$ at concentrations spanning $10^6$ to $10^9$ $cm^{-3}$, FT-1 is able to nucleate and monitor particles from precursor gases at a variety of conditions. After particles are nucleated in FT-1 where size distributions are recorded, particles are then transported to the growth tube (FT-2), where the particles undergo further growth at room temperature for a longer residence time (4 min) with the ability to precisely control other trace gases in the growth environment. According to our knowledge, this is the first flow tube experimental setup that allows for aerosol nucleation and growth processes to be investigated independently but at the same time.

The experimental setup of the nucleation region (FT-1) was based on (Benson et al., 2009;Benson et al., 2008;Erupe et al., 2011;Young et al., 2008b;Yu et al., 2012;Yu et al., 2017a;Benson et al., 2011). It consists a photolysis region where $H_2SO_4$ is generated photochemically and monitored followed by a temperature-controlled nucleation tube. In the photolysis region, OH radicals were produced via photodissociation of water vapor in a quartz tube using a mercury lamp (Pen-Ray Model 11SC-1). The mercury lamp was located in a temperature-controlled enclosure filled with a constant nitrogen flow. UV intensity was adjusted with an aperture over the slit in the enclosure exposing the quartz tube to the UV lamp. UV intensity was monitored with a CsI phototube (Hamamatsu Model R5764) and picoammeter (Keithley 6732). Measurements of UV intensity were taken to ensure consistency between experimental trials. $SO_2$, $O_2$ and $N_2$ gases were introduced to the flow tube immediately after the photolysis region. $H_2SO_4$ forms from the $SO_2 + OH$ reaction. Heating tape was applied to the $H_2SO_4$ production region to suppress nucleation prior to entering the temperature-controlled nucleation zone. The $H_2SO_4$ production region was monitored with a condensation particle counter (CPC, TSI 3776) and particle sizing magnifier (PSM, Airmodus A09) to ensure no particles were formed before the flow entered the nucleation region.

$H_2SO_4$ concentrations at the beginning of the nucleation tube were measured with a nitrate-based chemical ionization mass spectrometer (CIMS) based on (Eisele and Tanner, 1993) continuously during the experiments. Calibration of $[H_2SO_4]$ with the nitrate-CIMS was described previously by (Young et al., 2008a). The CIMS was operated with an inlet flow of 5.0 SLPM and an ion-molecule reaction time of 0.05 s. The lower limit of detection was calculated to be ~$1 \times 10^5$ $cm^{-3}$.

No base compounds were added, but base compounds were present in the flow tube as impurities likely generated from deionized water used for $H_2SO_4$ production and RH control (Erupe et al., 2011;Yu et al., 2012). $NH_3$ and amines were not measured during the 2018 IOP, but they were measured under very similar experimental conditions during the entire 2017 IOP with an ethanol-CIMS at the beginning of the nucleation tube (You et al., 2014;Yu and Lee, 2012 ). Detection limits of

NH$_3$/amines in our CIMS were pptv or sub-pptv with a 1-min integration, as previously discussed elsewhere (Benson et al., 2010;Erupe et al., 2011;You et al., 2014;Yu and Lee, 2012 ).

The nucleation tube is an 80 cm long Pyrex glass tube with an i.d. of 4.85 cm. The temperature of the nucleation tube was controlled with a circulating bath and a water-based potassium formate heat transfer fluid (Dynalene HC-50, Dynalene, Inc.) to adjust the temperature between 258 and 297 K. RH was adjusted by directing some of the dry nitrogen makeup flow through deionized water in a water bubbler. Thus, in our experimental setup, changes in RH in the nucleation tube did not affect the OH radical concentrations in the photolysis region. Temperature and RH probes (CS-215, Campbell Scientific) were used to monitor the conditions at the beginning of the photolysis region, as well as at the end of the nucleation region. An additional temperature and RH probe (Traceable, Fisher Scientific) was applied inside the nucleation tube to confirm the RH. Residence time in the nucleation region (FT-1) was 45 s.

Particle concentrations at the exit of FT-1 were measured with a PSM (Vanhanen et al., 2011). The PSM saturator flow was operated with a 240-step cycle between 0.1 - 0.9 SLPM at a rate of 1 s per step, giving saturator flow dependent cut-off sizes between 1.26 nm and 2.85 nm. These cut-offs were resolved to six size bins in an inversion method based on (Lehtipalo et al., 2014) producing size distributions with six size bins: 1.26 - 1.53 nm, 1.53 - 1.79 nm, 1.79 - 2.06 nm, 2.06 - 2.32 nm, 2.32 - 2.59 nm, and 2.59 - 2.85 nm. Particle concentrations were also monitored with a scanning mobility particle sizer (SMPS) consisting of a differential mobility analyser (DMA, TSI 3080) and a CPC (TSI 3776). However, under the typical experimental conditions, particles above 3 nm in diameter did not appear even with the most favourable conditions for nucleation and growth (e.g., high [H$_2$SO$_4$], high RH and low temperature).

During the experiments, [H$_2$SO$_4$] was varied by adjusting the aperture on the mercury lamp housing (hence varying [OH]) at a fixed [SO$_2$], allowing for a range of [H$_2$SO$_4$] spanning roughly one order of magnitude for a given dilution of SO$_2$. [H$_2$SO$_4$] was further varied by adjusting the SO$_2$ dilution, allowing for measurements spanning [H$_2$SO$_4$] of $10^6$ to $10^9$ cm$^{-3}$. The PSM measurements showed that each experimental condition was "stabilized" typically after ~30 min for a specific set of [H$_2$SO$_4$], RH and temperature.

The photolysis and nucleation tubes were cleaned thoroughly with deionized water, citric acid solution and ethanol and allowed to dry for 24 hours while heated to 60 $^o$C with pure N$_2$ flowing through the flow tube. Between experiments, the photolysis and nucleation tubes were continuously flushed with dry vaporized liquid nitrogen. A constant flow of N$_2$ was passed through the experimental apparatus at all times during the IOP to ensure that the conditions inside the tube would remain constant and there would be no intrusion of room air.

The nucleated clusters (smaller than 2 nm) were transported to the growth tube (FT-2) for further growth with an extended residence time (4 min). FT-1 and FT-2 were coupled with an 8-inch stainless steel tube with additional inlet ports for injection of ozone, zero air and SO$_2$. The growth tube was described by (Krasnomowitz et al., 2019;Stangl et al., 2019). The growth tube consists of a 1.52 m long and 0.2 m i.d. fused quartz tube fitted with stainless steel funnels on each end that reduce the i.d. down to 0.051 m. The total volume of the tube and entrance and exit funnels is 52.4 L, giving a surface-to-volume ratio of 0.24 cm$^{-1}$. The 8-inch straight tube fitting allows carrier/reactant gases to enter the tube via an axial inlet port and continuous

flow through the entire length of the reactor during the course of an experiment. The end of the tube was attached to an ozone monitor (Thermo Scientific 49i), a hygrometer (Traceable, Fisher Scientific), and an SMPS (TSI 3938, 3788).

FT-2 has several ports at the inlet to inject additional gases in to the system to observe their effects on further growth of freshly nucleated particles. Ozone was added to FT-2 using the calibration ozone generator on a Thermo Scientific 49i. Ozone was
5 varied in FT-2 by adjusting the UV intensity of the calibration lamp. The residence time in FT-2 could be changed by varying the flow of zero air in to FT-2. During a typical experimental run using both FT-1 and FT-2, conditions would only be changed in FT-2, with temperature, RH and precursor species in FT-1 held constant throughout the entire experiment. The experiments undertaken in this study measured two effects in the system: the effect of changing temperature in the nucleation region as well as the effect of varying ozone (in the co-presence of $SO_2$) in the growth region.

**2.2 Calculations of Nucleation (J) and Growth Rate (GR)**

Calculations of $J$ were made based on (Yu et al., 2017a). Briefly, $J$ was calculated according to the following approximation:
$$J_0 \approx N_{tot} \times nk_L, \tag{2}$$
where $J_0$ represents the nucleation rate corresponding to the initial sulfuric acid concentration ($[H_2SO_4]_0$) measured at the beginning of the nucleation tube, $N_{tot}$ the total number concentration of particles detected at the end of the nucleation region,
$n$ the nucleation theorem power and $k_L$ the diffusion-limited, pseudo first-order wall loss coefficient (Hanson and Eisele, 2000). The nucleation theorem power $n$ was experimentally determined for each set of experiments by the linear fit of Log $N_{tot}$ vs. Log $[H_2SO_4]_0$, Our $k_L$ was typically 0.01 s$^{-1}$. Eq. 2 allows us to obtain $J_0$ at different $[H_2SO_4]_0$ using $n$ and $N_{tot}$ of the formed clusters (all smaller than 2 nm in the present study).

To calculate $GR$, the critical cluster size was determined experimentally, with the critical cluster size corresponding to the y-
20 intercept of the linear fit between the mean particle diameter, $D_p$, and the $[H_2SO_4]_0$ (e.g., Fig. 3). $D_p$ was obtained using the inversion of the PSM size distribution measured at the end of FT-1. $GR$ was calculated by the difference between the critical size and $D_p$ divided by the nucleation time.

The growth rate factor $k_G$ is defined as the growth rate enhancement over 1 pptv $H_2SO_4$ causing 1 nm hr$^{-1}$ of growth. The collision limited condensation of $H_2SO_4$ of 1 pptv contributes roughly 1 nm hr$^{-1}$ of growth rate at a temperature of 298 K
(Nieminen et al., 2011). Thus, the $k_G$ is an indicator of the deviation of the actual growth rate compared to this conventional collision-limited $GR$. The $k_G$ were determined by the expression derived by (Yu et al., 2017a):
$$k_G = \frac{\Delta D_{p,tr} \times 10^7 cm^{-3}}{[H_2SO_4]_0} \frac{k_L}{1-e^{-nk_L t_r}}, \tag{3}$$

Where $\Delta D_{p,tr}$ represents the change in diameter over the residence time ($t_r$) in the nucleation tube (FT-1). $\frac{\Delta D_{p,tr}}{[H_2SO_4]_0}$ was experimentally determined from the slope of the $D_p$ vs. $[H_2SO_4]_0$ plot (e.g., Fig. 3). $\frac{k_L \times 10^7 \, cm^{-3}}{1-e^{-nk_L t_r}}$ can be calculated from $n$ and
30 $k_l$ values. The $k_G$ is the product of this slope and $\frac{k_L \times 10^7 \, cm^{-3}}{1-e^{-nk_L t_r}}$. Extrapolation to the y-axis of Figure 3 gives a value for the

critical cluster diameter. $\Delta D_{p,tr} = 0$ for the critical cluster and thus $D_{p,\ critical}$ will be equal to the y-intercept of the linear fit of each data set.

## 2.3 Uncertainty analysis

The uncertainties in the $J_0$ calculation arise from uncertainties in CIMS $H_2SO_4$ measurements, measurements of particle concentrations in the PSM and the size inversion from PSM-measured number concentrations. The uncertainty in the $H_2SO_4$ CIMS is approximately 60% (Erupe et al., 2010;Eisele and Tanner, 1993;Benson et al., 2008;Benson et al., 2009;Kürten et al., 2012;Petäjä et al., 2009). The wall loss calculation has ±7% uncertainty (Hanson and Eisele, 2000). The measurement error in the PSM size distribution, based on the standard error calculated between run-to-run experiments under an identical condition, is ±26%. The uncertainties in inversion of the particle diameter from the PSM measurement are estimated around 12% (or ±0.2 nm) for inorganic particles in the size range observed in this experiment (Lehtipalo et al., 2014;Lehtipalo et al., 2016;Kulmala et al., 2013;Yu et al., 2017a). Propagation of the errors in sulfuric acid measurement, wall loss, and the PSM measurements results in an overall uncertainty of ±65.5% in the calculation of $J_0$.

## 3. Results and discussion

### 3.1. Nucleation and growth in FT-1

Table 1 shows the typical experimental conditions used in the FT-1 (nucleation tube) and FT-2 (growth tube) during the 2018 IOP study. In the nucleation tube, temperature was varied from 258 to 297 K and RH from 4% to 85%. $[H_2SO_4]$ spanned from $10^6$ to $10^8$ cm$^{-3}$, corresponding to RA of $10^{-5}$ to $10^{-2}$. RA was calculated using the sulfuric acid saturation vapor pressures provided by (Vehkamaki et al., 2002). $NH_3$ and amine measurements not taken during this study, but they were undertaken during the 2017 IOP campaign using the same experimental apparatus, precursor gases and cleaning schedule as the 2018 campaign that is the concern of this study. The CIMS-measured $NH_3$ (during the 2017 IOP) was $14.2 \pm 6.7$ ppt (Fig. 2). Thus, the ratio of $[NH_3]/[H_2SO_4]$ ranged from 0.6 to 268. According to (Schobesberger et al., 2015) and (Dunne et al., 2016), these ratios represent ternary nucleation and some nucleation in a transition regime between binary and ternary, when considering only the effects of $NH_3$ (without amines). Overall, the measured nucleation rates ranged from 10 - $10^5$ cm$^{-3}$ at higher temperatures and lower $[H_2SO_4]$ up to $10^5$ cm$^{-3}$ at lower temperatures and higher $[H_2SO_4]$. The observed $GR$ ranged from 1 to 80 nm h$^{-1}$. The growth tube (FT-2) was kept at room temperature (297 K) and dry conditions (RH of 10%). $SO_2$ was added in the range from 100 ppbv to 5 ppmv and ozone from 0 to 248 ppbv.

Figure 3 shows the measured $D_p$ with the PSM at the end of FT-1 as a function of the initial $[H_2SO_4]_0$ at the temperature between 258 and 297 K. $[H_2SO_4]_0$ was varied from $8\times10^6$ cm$^{-3}$ to $7\times10^7$ cm$^{-3}$. The RH was kept in relatively narrow range between 20% and 30%. The y-intercept in Fig. 3 indicates the critical cluster diameter was estimated to be between 1.6-1.7 nm depending on temperature, with lower temperatures resulting in smaller critical cluster diameters. However, since the PSM

inversion that determines $D_p$ has an uncertainty of $\pm 0.2$ nm, it is difficult to discuss the temperature trends of the critical cluster diameter. This critical size is consistent with (Kulmala et al., 2013) and (Almeida et al., 2013), which determined critical cluster diameters of $1.5 \pm 0.3$ nm and 1.7 nm, respectively. For a given $[H_2SO_4]_0$, the mean $D_p$ at the end of the 45 second residence time in FT-1 was larger for lower temperatures. Previously, (Glasoe et al., 2015;Yu et al., 2017a) have also shown increasing $GR$ with increasing $[H_2SO_4]$ from flow tube experiments. The slope of $D_p$ vs. $[H_2SO_4]_0$ increased with each 10 degree decrease in temperature over the course of these experiments. Thus, the growth rate factor $k_G$ also increased with subsequent temperature decreases (e.g., from 1.27 at 297 K to 12.6 at 258 K). These results indicate that lower temperatures promote the faster growth of particles due to the reduction in saturation vapor pressures of $H_2SO_4$ at lower temperatures.

Figure 4 shows the relationship of Log $J$ vs. Log RA for different temperatures. Experiments were conducted at 10 K intervals, starting from 297 K down to 258 K for RH between 41% and 45%. Across all temperature and RH experiments conducted, in general, $J$ values were shifted 2 to 3 orders of magnitude above previous literature values of flow tube nucleation studies (Brus et al., 2010;Yu et al., 2017a) and 4 to 5 orders of magnitude above CLOUD measurements of $H_2SO_4$ nucleation as shown in the figure using CLOUD data from (Dunne et al., 2016). This upward shift was consistent across trials. Based on our measured $NH_3$ and amine concentrations (Fig. 2), this upward shift is consistent with the nucleation rate enhancement due to synergistic effects of $NH_3$ concentrations on the order of 20 to 30 pptv and dimethylamine concentrations on the order of 5 ppt reported by other studies (Glasoe et al., 2015). There was a consistent relationship between Log $J$ and Log RA. Except for 288 K and 297 K, where the slope of Log $J$ vs. Log RA was approximately 2, slope was 3 for all trials, with the best-fit lines shifting towards higher values of RA as temperature decreased. Hanson and colleagues provided comprehensive analysis of Log $J$ vs. Log RA (or Log $[H_2SO_4]$) obtained from flow tube studies (Glasoe et al., 2015;Zollner et al., 2012). In general, flow tube studies from various groups have shown slopes between 3-6 for the ternary system (Glasoe et al., 2015;Zollner et al., 2012;Brus et al., 2010;Berndt et al., 2014;Erupe et al., 2011;Yu et al., 2012;Hanson et al., 2017). Using the CLOUD experiments, (Dunne et al., 2016;Almeida et al., 2013;Kirkby et al., 2011) also showed the slope of 3 for the ternary system. This slope is consistent with the base-stabilization mechanism provided by (Chen et al., 2012;Jen et al., 2014;Jen et al., 2016) that the bottleneck clusters contain 3-4 $H_2SO_4$ molecules with at least one base molecule. It was previously believed that the slope of Log $J$ vs. Log RA dictates the amount of $H_2SO_4$ molecules present in the critical cluster based on classical nucleation theory (CNT) (Kashchiev, 1982;McGraw and Zhang, 2008), which then would imply here that the critical cluster contains three $H_2SO_4$ molecules for the ternary system. However, more recent work by (Malila et al., 2013;Vehkamäki et al., 2012) has shown that this conclusion may be an oversimplification of the mechanism of particle formation resulting from an application of CNT with an incomplete understanding of the free energy maxima and minima. As a result, there is caveat when the critical cluster composition is determined by the simple relationship between $J$ and $[H_2SO_4]$.

Figure 5 shows the measured $J$ as a function of temperature for $[H_2SO_4]$ between $2 \times 10^7$ and $3 \times 10^7$ cm$^{-3}$ and RH between 15% to 45%. $J$ increased with the decreasing temperature in the temperature range above 268 K. The higher $J$ at lower temperatures is consistent with predictions from CNT (Seinfeld and Pandis, 2016). However, a shift in slope is visible around 268 K, indicating that the dependence of $J$ on temperature becomes less significant at low temperatures. The variation seen across RH

at higher temperatures also becomes negligible below this temperature. Thus, these results indicate that at temperatures below 268 K, Gibbs free energy barriers are reduced significantly. This is consistent with (Duplissy et al., 2016) found barrierless particle formation at lower temperatures. As discussed above, the base contamination present in the conditions for this study resulted in elevated $J$ values, which also resulted in this barrierless kinetic nucleation occurring at a relatively higher temperature.

### 3.2. TANGENT experiments: Further growth of clusters in FT-2:

In order to determine whether newly formed particles nucleated at lower temperatures can survive downward transport to warmer temperature conditions, we conducted two tests (Figs. 6 & 7): the first test had a much lower temperature in FT-1 (268 K) than in FT-2 (297 K) and the second test had the same temperature (297 K) in both tubes. During the first test (Fig. 6), the average particle concentration coming out of FT-1 was $2\times10^5$ cm$^{-3}$ with a median diameter of 1.9 nm at $[H_2SO_4]_0$ of $6\times10^7$ cm$^{-3}$ and RH of 10%. These newly formed particles were further mixed with an additional zero air flow at a 1:6 dilution. The $[SO_2]$ was 83 ppbv and the ozone level was varied from 0 to 248 ppbv in FT-2. No particles were observed coming out of FT-2 when ozone was absent, indicating that $SO_2$ alone does not cause nucleation and growth of clusters. However, in the presence of ozone and $SO_2$, continuous nucleation and further growth of transported clusters took place in FT-2. The particle concentration measured at the end of FT-2 was closely correlated with the ozone concentration: the particle concentration ranged from $3\times10^2$ cm$^{-3}$ (with $D_p$ of 2.8 nm) at the lowest ozone of 28 ppbv up to $5\times10^4$ cm$^{-3}$ ($D_p$ of 3.4 nm) at the maximum ozone of 248 ppbv. Thus, the particle concentration in FT-2 at the highest ozone was even greater than that coming out of FT-1 after dilution, indicating that a high ozone load resulted in additional nucleation and the further growth of clusters in FT-2. This could be the result of the remaining $H_2SO_4$ vapour passing through to FT-2, but we excluded this possibility. After considering wall loss in FT-1 and the 1:6 dilution FT-2, $[H_2SO_4]$ in FT-2 was estimated to be $1.15\times10^6$ cm$^{-3}$, which can result in $J$ only on the order of $10^1$ cm$^{-3}$ s$^{-1}$ at room temperature and the dry condition, as shown from the results obtained in FT-1 (e.g., Fig. 4). However, the measured formation rate ($J$) of particles in FT-2 was 190 cm$^{-3}$s$^{-1}$, in contrast to this estimation. There was another possibility that ozone reacted with possible organic impurities in FT-2 to produce OH, which oxidized $SO_2$ to produce $H_2SO_4$ and nucleated in FT-2, but no organics were added in our experiments. It is not clear at present what is the cause of nucleation in FT-2 and this requires future study. However, it was clear that the co-presence of ozone and $SO_2$ was an important factor in preventing evaporation of newly formed particles and facilitated them to the further grow at the higher temperature. The overall $GR$ of particles in FT-2 was from 14.9 to 23.1 nm h$^{-1}$, depending on the ozone concentration. These results may imply some heterogeneous reactions occurring on acidic sulfuric acid clusters, in a similar way to form sulphate from oxidation reactions of $SO_2$ on acidic particles as proposed by (Hung and Hoffmann, 2015).

The second test was conducted with FT-1 and FT-2 both at a constant temperature of 297 K (Fig.7). FT-1 was held at a constant $[H_2SO_4]_0$ of $1.3\times10^8$ cm$^{-3}$. The total particle concentration out of FT-1 for this experiment was $1.7\times10^5$ cm$^{-3}$ with a mean diameter of 1.9 nm. Aside from the higher $[H_2SO_4]_0$, which was necessary to produce the same particle concentration in the higher temperature FT-1, particle count and mean diameter were similar to the experiment with a cold FT-1. However, the size

distributions in Figure 6a and Figure 7a indicate that there were more particles in the higher diameter size bins when the temperature in FT-1 was lower. At the end of FT-2, particle concentration and size were again closely correlated with ozone concentration, ranging from $8.6\times10^1$ cm$^{-3}$ ($D_p$ = 2.7 nm) at the lowest ozone concentration up to $2.3\times10^4$ cm$^{-3}$ ($D_p$ = 3.8 nm) at the highest ozone load. These concentrations and sizes result in a growth rates ranging from 12.0 to 28.1 nm hr$^{-1}$, indicating that the growth the highest ozone concentrations was faster during the constant temperature experiment at. This was due to a greater concentration of smaller particles formed at higher [H$_2$SO$_4$]$_0$ in the second test.

Examining the contrast between Figure 6, we see notable effects of temperature gradient on the survival of particles to the end of FT-2. While the average size at the end of the nucleation tube was similar in each experiment ($D_p$ = 1.9 nm with 268 K FT-1, $D_p$ = 1.9 nm with 298 K FT-1), we can see from the size distributions in each experiment that the low-temperature FT-1 resulted in a greater concentration of particles in the larger size bins. These particles (once they are stabilized, e.g. by base compounds) are more thermodynamically stable and therefore have a greater ability to survive evaporation when transported to the warmer environment of FT-2. Particles formed in warmer environments have a greater energy barrier to overcome before they become stable enough to grow spontaneously (Lovejoy et al., 2004).

The additional nucleation occurring in FT-2 in the co-presence of SO$_2$ and ozone cannot be explained by the current knowledge. These results suggest some possible heterogeneous reactions involving SO$_2$ and ozone on acidic clusters. One possibility is the presence of alkaline species such as transition metal or high NH$_3$/amine concentrations that could increase the pH of the particles, which can contribute to the production H$_2$SO$_4$ by SO$_2$ oxidation in the presence of ozone (Seinfeld and Pandis, 2016;Hung and Hoffmann, 2015). However, CLOUD studies have often found that particles nucleated in the presence of NH$_3$/amines remain acidic (Lawler et al., 2016). Understanding how SO$_2$ and ozone interact each other on the acidic nanoparticles to facilitate nucleation and growth will require future studies.

We also examined the additional effects of organics and base compounds on $GR$ in FT-2. First, to see the effects of possible impurities of organics, we used the growth rate parameterization from (Tröstl et al., 2016):

$$GR = kD_p[HOM]^p \tag{4}$$

Solving Eq. 4 for [HOM] using our observed growth rates of 12.0 to 28.1 nm hr$^{-1}$, $D_p$ = 1.9 nm, k = $5.2\times10^{-11}$ and p = 1.424 (Tröstl et al., 2016) results in a [HOM] of 2.8 to 3.8 pptv necessary to account for these growth rates. The amount of monoterpene required to produce that amount of HOMs was calculated using equation 16 from Trostl et al., 2016 solved for [MT]:

$$[MT] = \frac{[HOM]*CS}{Y_1 k_1 [O_3]} \tag{5}$$

Using the yield of HOMs from ozonolysis of monoterpenes $Y_1$ = 2.9% (Kirkby et al., 2016), temperature dependent reaction rate of ozone and α-pinene $k_1$ = $8.06\times10^{-17}$ (Khamaganov and Hites, 2001), and $10^{-3}$ for condensation sink ($CS$, calculated from size distributions in FT-2) it would require between 100 and 1000 ppbv of monoterpenes to account for that concentration of HOMs. These high concentrations were very unlikely to be present in FT-2 considering the rigorous cleaning of the TANGENT apparatus as well as the lack of added monoterpenes to the system.

Second, to see the possible multicomponent growth due to the presence of ammonia in the system, we utilized the parameterization of multicomponent growth from equation (4) of (Lehtipalo et al., 2018):

$$GR = k_1[H_2SO_4]^a + k_2[H_2SO_4]^b[NH_3]^c + k_3[Org]^d \qquad (6)$$

Where $k_1$ and $k_2$ are $2.05\times10^{-7}$ and $6.69\times10^{-11}$, respectively (Lehtipalo et al., 2018) and $a = b = c = d = 1$. Assuming $[Org] = 0$ for simplicity, and $[H_2SO_4]$ and $[NH_3]$ values from the temperature gradient experiment ($1.15\times10^6$ cm$^{-3}$ H$_2$SO$_4$, 14 ppt NH$_3$ estimated from 2017 IOP measurements), we should expect a $GR$ of 0.24 nm hr$^{-1}$. However, this calculation only considers the presence of NH$_3$ in the system, when it is very likely that there were amines present at pptv levels (Fig. 2). (Glasoe et al., 2015) and (Yu et al., 2012) found synergistic effects on $GR$ when both NH$_3$ and methylamine/dimethylamine are present that can enhance $GR$ by 1 to 2 orders of magnitude, which would put these calculated $GR$ values in range of the observed 23.1 nm hr$^{-1}$ $GR$ seen in the FT-2.

Also, cluster-cluster collision in the presence of stabilizing base compounds such as trace amines that were present in TANGENT can enhance growth by an order of magnitude (Lehtipalo et al., 2016). Thus, it seems that the observed high GRs in FT-2 were at least in some part due to multicomponent effects of base compounds and cluster-cluster collisional growth effects on $GR$, in addition to potential heterogeneous processes discussed above.

The chemical composition of the particles nucleated and grown in this experiment is an element that is likely to have an influence on their stability, growth, and survivability in a different environment. We did not measure chemical composition of clusters and particles in the present study. However, we can surmise the chemical composition of the particles in this study based on CLOUD studies that utilized APi-TOF instruments (Duplissy et al., 2016;Kirkby et al., 2011;Kürten et al., 2014). These studies, conducted in similar precursor conditions of H$_2$SO$_4$ and NH$_3$/amines at pptv levels, found that pure H$_2$SO$_4$ clusters did not exist beyond dimers and trimers at the lowest temperatures; NH$_3$ or amines particles consisting of H$_2$SO$_4$ molecules clustered with ammonia molecules at ratios up to 1:1 were common. It thus can be expected that the particles in this study had a similar chemical composition, and the presence of bases in these newly formed particles was a critical factor for their survival and further growth. This chemical composition is also consistent with above analysis about the $GR$ in the FT-2, derived from the multicomponent nucleation and growth processes by (Lehtipalo et al., 2018).

Our results in the TANGENT setup using two different temperatures in FT-1 and FT-2 show that particles were observed at the end of the room temperature nucleation tube after they were initially nucleated at lower temperatures growth tube, which is similar to the experiments in which FT-1 was held at cold temperatures for particle formation before particles were transported and allowed to grow further in the warmer environment of FT-2. These results thus clearly demonstrate that indeed small clusters can survive evaporation when transported within different temperature regions. This is consistent with the mechanism of particle formation and downward transport in the marine boundary layer, where nucleation takes place by sulfuric acid formed in the free troposphere at cooler high altitudes before they are transported downwards to warmer conditions (Russell et al., 1994;Raes, 1995;Clarke, 1993). Our results can also help to explain the presence of newly formed particles observed in Amazon forests by (Wang et al., 2016), which concluded that the particle loads observed in the boundary layer could be the result of downward transport of particles formed in the colder free troposphere.

Also, these high *GR* and additional nucleation observed in FT-2 in the co-presence of $SO_2$ and ozone may be relevant to the frequent and strong NPF events observed in extremely polluted Chinese mega cities (Guo et al., 2014;Yao et al., 2018;Yu et al., 2017b). In these environments, there exists a high pre-existing particle load that would ordinarily suppress NPF by acting as condensation sink of newly formed particles. However, in these highly polluted areas NPF events are observed with some regularity, which cannot be explained by the current knowledge.

## 4. Conclusions and Implications

We have conducted experiments to study the temperature dependence of aerosol nucleation and growth using the TANGENT setup. This setup consists of two flow tubes which enable us to study nucleation and subsequent growth independently. In the nucleation tube, temperature was varied from 258 to 297 K and RH from 4% to 85%. $[H_2SO_4]$ spanned $10^6$ to $10^8$ $cm^{-3}$, which corresponds to RA of $10^{-5}$ to $10^{-2}$. Based on the measured $[NH_3]$ to $[H_2SO_4]$ ratios, it was most likely that nucleation took place via the ternary process. The growth tube was kept at room temperature and the dry condition (RH of 10%). $SO_2$ was present at 100 ppbv to 5 ppmv and ozone at 0 to 248 ppbv.

Our results indicate that lower temperatures enhance both nucleation and growth rates as predicted by CNT. However, the temperature effects on nucleation rates become less important at lower temperatures (below 268 K), consistent with CLOUD studies (Duplissy et al., 2016) which found that sulfuric acid nucleation takes place at the kinetic limit without a Gibbs free energy barrier when temperature was low. These results emphasize the importance of $NH_3$ and other ternary species at warmer temperatures, for example, especially in the conditions present in the boundary layer.

Our results demonstrate that clusters formed at lower temperatures, while being transported to warmer temperatures, can survive evaporation and even grow further in the presence of $SO_2$ and ozone. We are providing the first this laboratory experimental evidence that support the mechanism of downward transport of newly formed particles from the free troposphere to the marine boundary layer proposed by several field and modelling studies (Russell et al., 1994;Raes, 1995;Clarke, 1993). Similarly, based on our results, it is reasonable to conclude that the new particles formed in the free troposphere over the Amazon forest are transferred downward to the warmer surface to act as a reservoir of nuclei mode particles (Wang et al., 2016), though the mechanism of growth is different from the one studied here ($SO_2$ + ozone).

Our results also show that the further growth is strongly dependent on the ozone level, implying that some unknown heterogeneous reaction processes involving $SO_2$ and ozone on sulfuric clusters may play important roles in NPF. These results can open a new research avenue for future studies for better understanding the roles of heterogeneous reactions involving nanoparticles and the effects of $SO_2$ on the nanoparticle growth. At present, it is not known why NPF takes place with high frequency and strong magnitude in extremely polluted megacities in China under the conditions with exceedingly high loadings of pre-existing aerosol particles (Guo et al., 2014;Yao et al., 2018;Yu et al., 2017b). The heterogeneous reactions involving $SO_2$ on nanoparticles proposed here may provide some key insights into understanding the frequent nucleation and fast growth observed in these regions, where there are also very high concentrations of $SO_2$ and ozone.

**Data Availability**

All data of this work can be obtained from Lee Tiszenkel (lt0021@uah.edu) and Shanhu Lee (shanhu.lee@uah.edu).

**Author Contributions**

SL and MJ designed the experiments and LT, CS, JK, QO and MA carried them out. LT, QO and HY developed code used in the data analysis. LT, CS and QO performed the data analysis. LT and SL prepared the manuscript with contributions from all co-authors.

**Acknowledgements**

This work was supported by NSF Awards AGS-1649719 and AGS-1649694.

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

**Table 1. Typical experimental conditions used in FT-1 and FT-2. *J* in FT-1 is the nucleation rate for sub-2 nm particles. *J* in FT-2 is the formation rate of total particles in the size range from 3 to 60 nm; the particle mean diameter ranged from 2.7 to 3.4 nm depending on the ozone concentration. *GR* in FT-1 is the growth rate of sub-3 nm particles. *GR* in FT-2 is the growth rate of total particles from 3 to 60 nm. $NH_3$ and amine measurements are from IOP 2017 under similar conditions.**

| FT-1: Nucleation Region | | | | | | FT-2: Growth Region | |
|---|---|---|---|---|---|---|---|
| Temperature (K) | 297 | 288 | 278 | 268 | 258 | Temperature (K) | 297 |
| RH | 20% - 45% | 8% - 60% | 12% - 80% | 23% - 80% | 46% - 85% | RH | 10% |
| $[H_2SO_4]$ $(cm^{-3})$ | $1\times10^7 - 3\times10^8$ | $7\times10^6 - 2\times10^8$ | $2\times10^6 - 7\times10^7$ | $2\times10^6 - 4\times10^7$ | $4\times10^6 - 7\times10^7$ | $SO_2$ (ppbv) | 100-5000 |
| RA | $4\times10^{-5} - 7\times10^{-4}$ | $7\times10^{-5} - 1\times10^{-3}$ | $6\times10^{-5} - 2\times10^{-3}$ | $3\times10^{-4} - 5\times10^{-3}$ | $3\times10^{-3} - 4\times10^{-2}$ | $O_3$ (ppbv) | 0 - 248 |
| GR (nm h$^{-1}$) | 1 - 20 | 1 - 20 | 2 - 80 | 3 - 45 | 2 – 35 | GR (nm h$^{-1}$) | 12.0- 28.1 |
| J (cm$^{-3}$ s$^{-1}$) | $10^1 - 10^5$ | $10^2 - 10^5$ | $10^2 - 10^5$ | $10^2 - 10^5$ | $10^2 - 10^5$ | J (cm$^{-3}$ s$^{-1}$) | 0 - 189.9 |
| $[NH_3]$ (pptv) | 14.2 ± 6.9 | | | | | | |
| $[NH_3]/[H_2SO_4]$ | 0.6 – 52.7 | 0.9 – 75.2 | 2.6 - 263 | 4.5 - 263 | 2.6 – 132 | | |
| [C1 amine] (pptv) | 4.5 ± 2.60 | | | | | | |
| [C2 amine] (pptv) | 44.8 ± 41.8 | | | | | | |
| [C3 amine] (pptv) | 7.27 ± 2.50 | | | | | | |
| [C4 amine] (pptv) | 21.7 ± 7.5 | | | | | | |
| [C5 amine] (pptv) | 13.9 ± 4.3 | | | | | | |
| [C6 amine] (pptv) | 8.42 ± 1.8 | | | | | | |

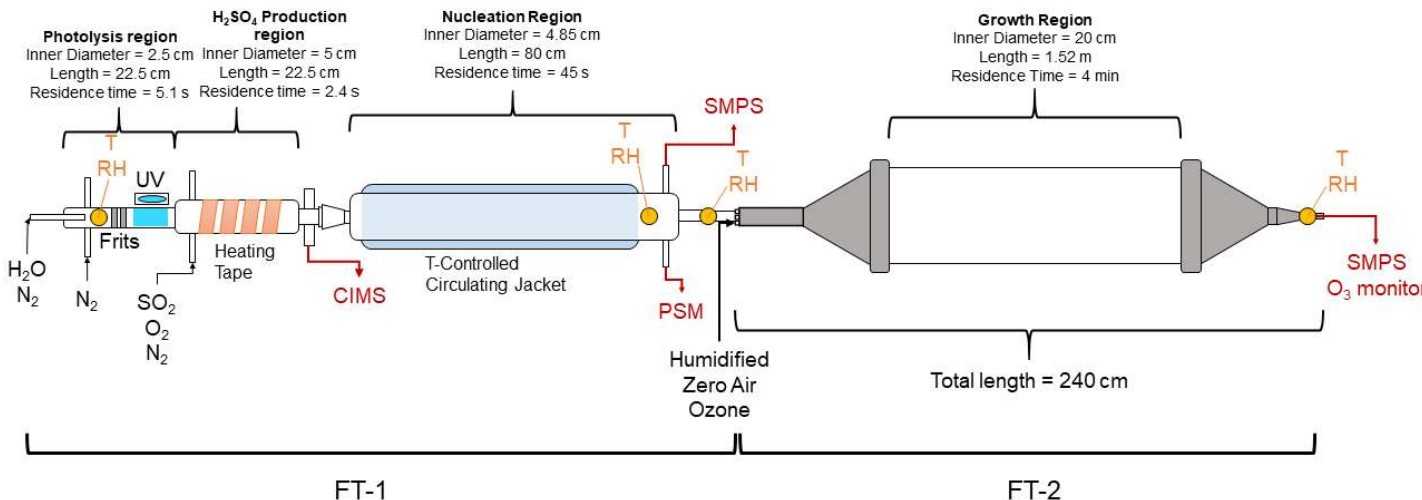

**Figure 1.** **Schematic diagram of the TANGENT experimental setup. This setup consists of two flow tubes (FT). T indicates temperature. FT-1 is used as the nucleation region and FT-2 as the growth region. Table 1 shows the typical experimental conditions used during the 2018 IOP study.**

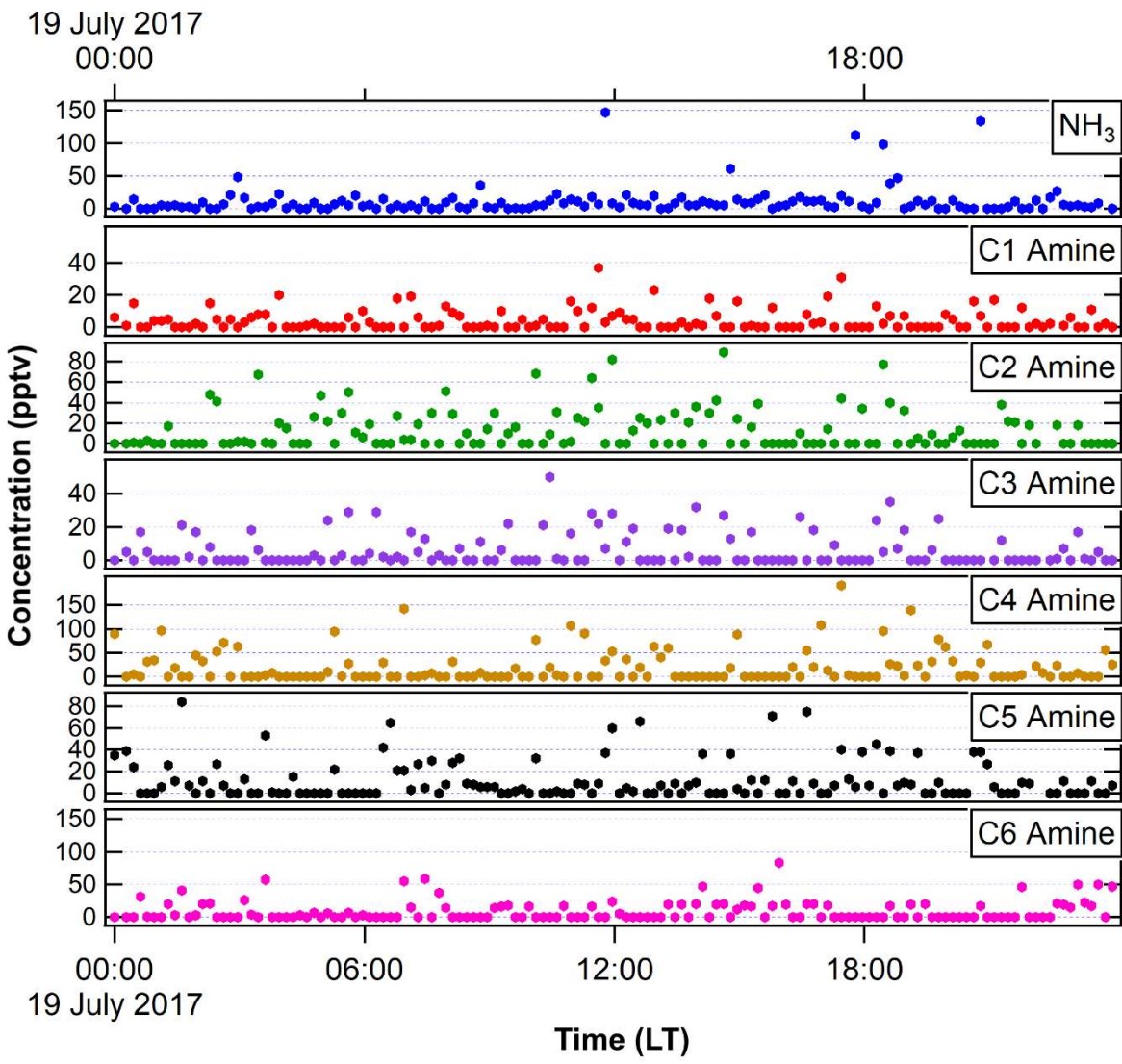

**Figure 2. NH₃ and amines measured with the ethanol-CIMS in the FT-1 during the 2017 IOP in a very similar experimental condition as in 2018 IOP. We show here an example of one day measurements (July 19, 2017).**

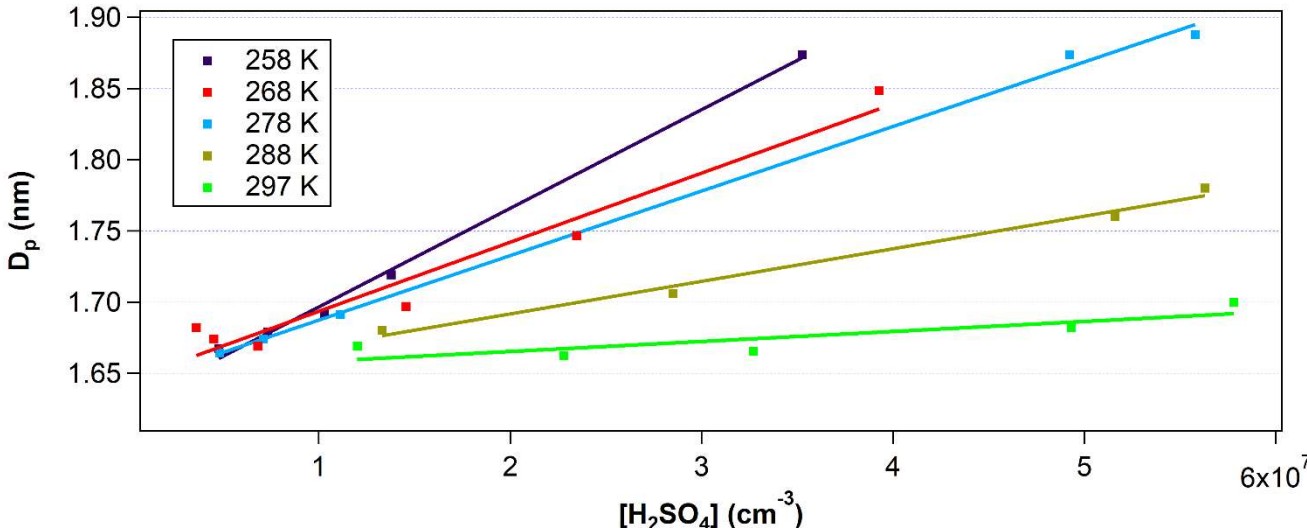

**Figure 3.** Mean diameter of particles ($D_p$) inverted from the PSM measurements at the end of the nucleation tube as a function of [$H_2SO_4$] and temperature. Data points were taken at RH between 20% and 30%. Solid lines are linear fittings of the measurement data (coloured squares) under different temperatures. Error in [$H_2SO_4$] is estimated to be ±60%. Error in $D_p$ at this size range is estimated to be ±0.2 nm, or approximately ±12%.

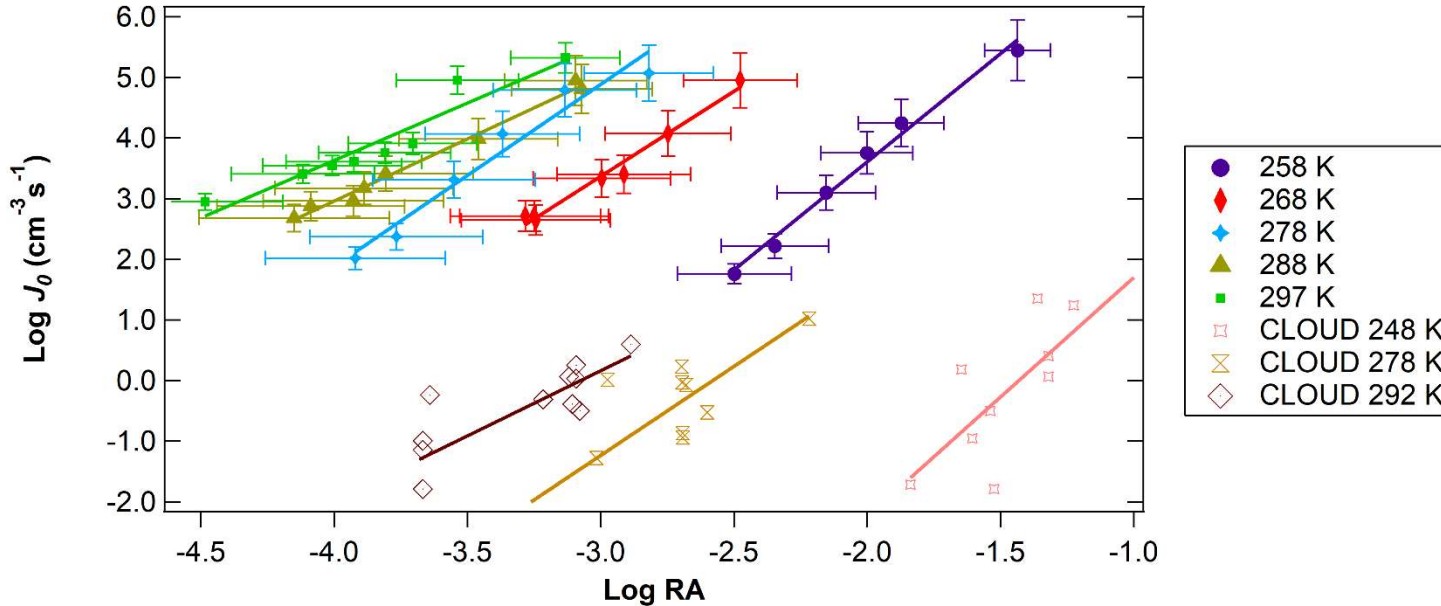

**Figure 4.** Log $J_0$ vs. Log RA for different temperatures at a relatively constant RH (41% - 45%). Temperatures ranged from 258 to 297 K. CLOUD data for similar RA and temperature conditions is shown from (Dunne et al., 2016)

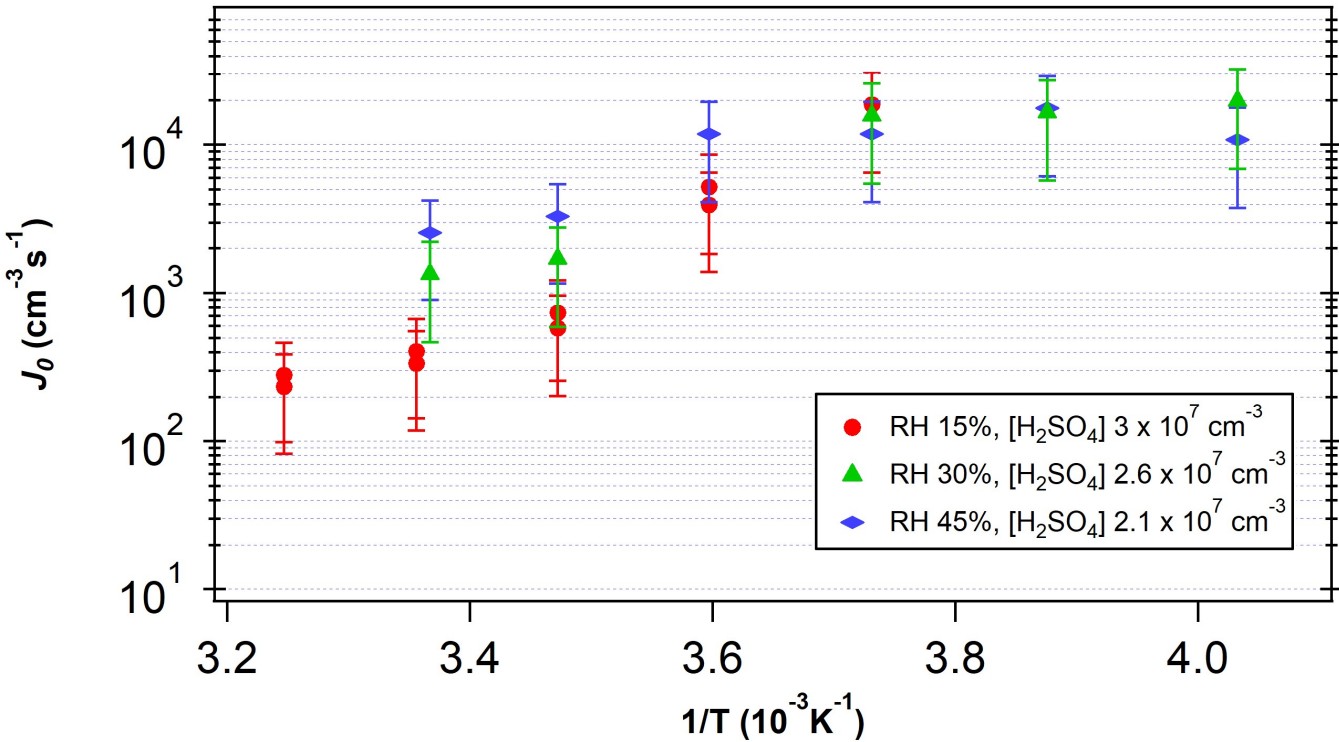

**Figure 5.** Log $J_0$ vs. 1/T for [H2SO4] between $2 \times 10^7$ and $3 \times 10^7$ cm$^{-3}$. T is the temperature in FT-1. RH ranged from 15% to 45%. Vertical bars indicate one standard deviation of the measured nucleation rates.

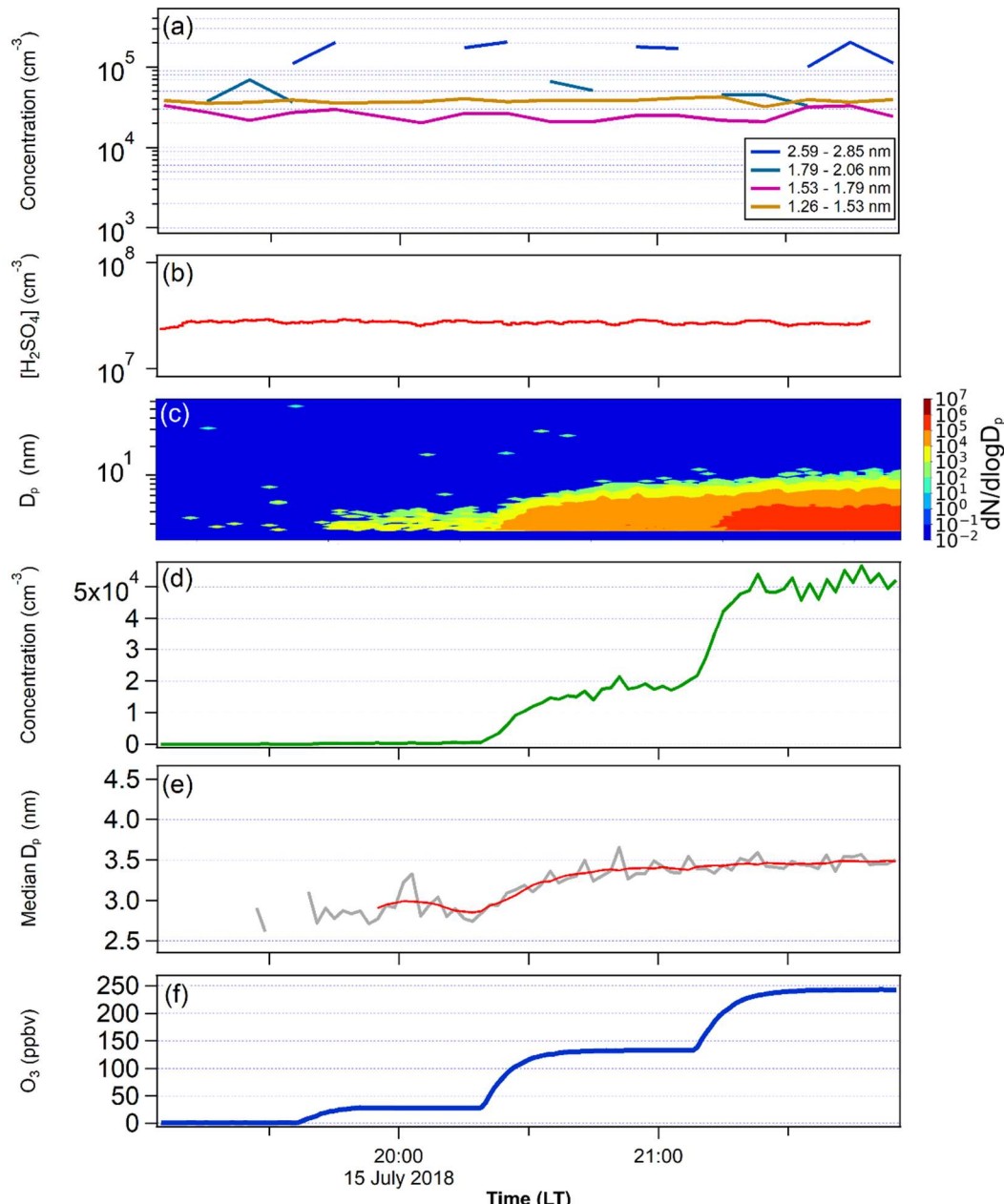

**Figure 6.** **(a) The PSM-inverted size distribution and (b) [H₂SO₄] measured in FT-1 during the temperature gradient TANGENT experiment.FT-1 was at 268 K, residence time 45 s. Total concentration at the end of FT-1 was $1.79\times10^5$ cm$^{-3}$ with a mean $D_p$ of 1.9 nm. (c) SMPS-measured particle size distribution, (d) total number concentration, (e) the particle median diameter $D_p$, and (f) Ozone concentrations in FT-2. FT-2 was kept at 297 K, and the residence time was 4 min. The red line in (e) indicates the average values of $D_p$. SO₂ was 500 and 83 ppbv in FT-1 and FT-2, respectively. H₂SO₄ was not measured at FT-2; however, after considering wall loss in FT-1 and the 1:6 dilution FT-2, [H₂SO₄] in FT-2 was estimated to be $1.15\times10^6$ cm$^{-3}$.**

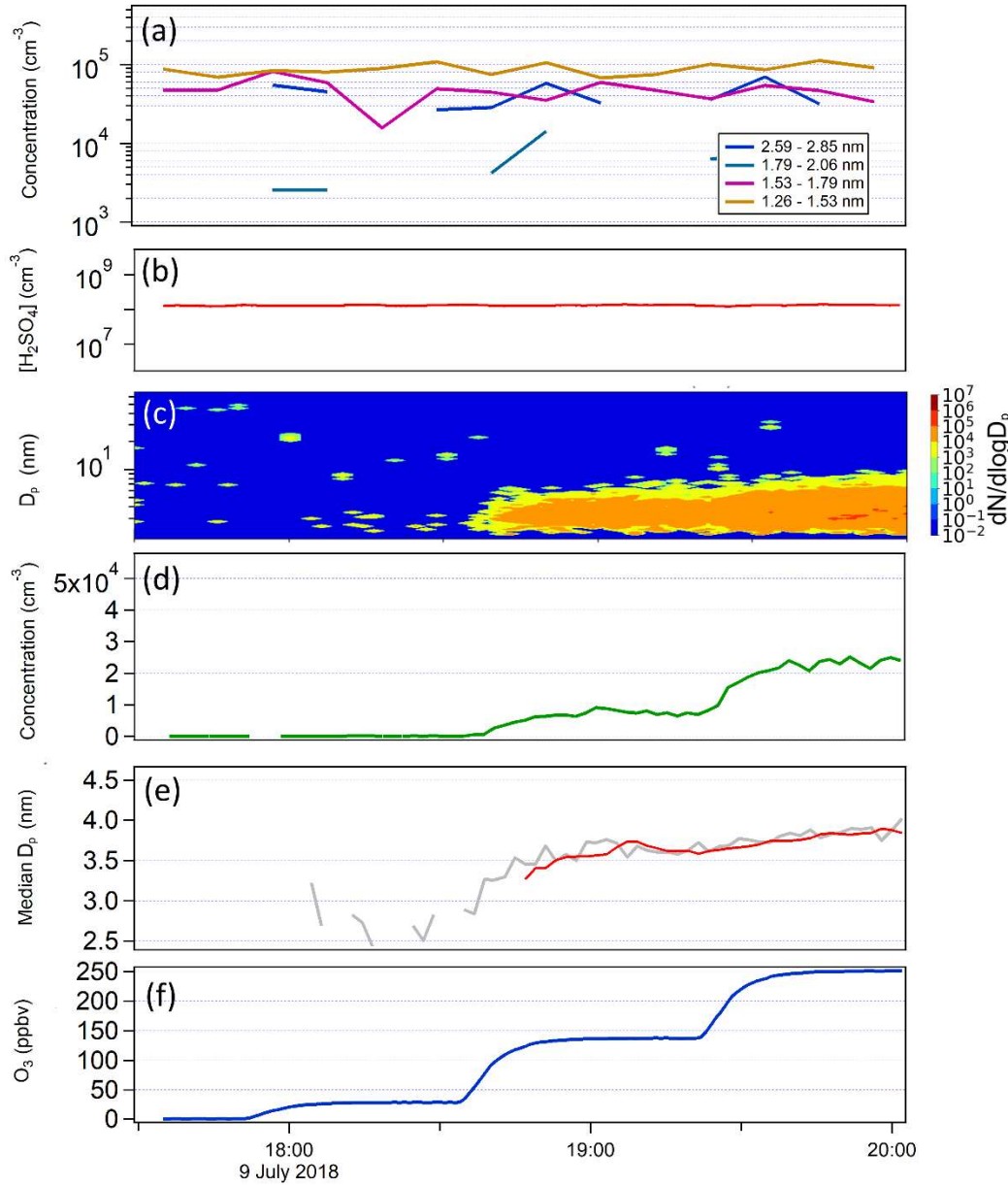

**Figure 7.** The same as Figure 6, except that FT-1 and FT-2 had a constant temperature in this test. Total particle concentration at the end of FT-1 was $1.7 \times 10^5$ cm$^{-3}$ with a mean $D_p$ of 1.9 nm and [H$_2$SO$_4$] in the FT-1 was $1.3 \times 10^8$ cm$^{-3}$. In FT-2, the [H$_2$SO$_4$] was estimated to be $2.5 \times 10^6$ cm$^{-3}$.