# Peer review of "Temperature Effects on Sulfuric Acid Aerosol Nucleation and Growth: Initial Results from the TANGENT Study"

_Atmospheric Chemistry and Physics, 2019_

## Referee Comment (RC1) · Anonymous Referee #1 · 3 Feb 2019

Review of Tiszenkel et al. Temperature effects on sulfuric acid aerosol nucleation and growth: Initial results from the TANGENT study.

The article is nicely written, relevant literature is cited and the shortcomings of the current study (mainly related to the experimental setup) are discussed appropriately.

However, I don't think the study brings anything new to the current scientific literature. The temperature effects on nucleation and early growth have already been discussed extensively in the previous study from partly the same author group (Yu et al. 2017), as well as in the studies from the CLOUD community. Given that the current study lacks crucial information about the cluster composition and measurements of base and

organic contaminations in their system (thus relying on many assumptions based on comparison to the previous literature), there is little value in the few additional data points and speculations that are presented.

Given their high nucleation and growth rates for each sulfuric acid concentrations in the nucleation tube, I agree with the authors, that the nucleation mechanism is most probably ternary, i.e. involving ammonia and/or amines. However, the authors have not measured the cluster composition at the end of the nucleation tube, which, most probably, critically affects the survival of the nucleated clusters in warmer temperatures (what they intended to study and which is presented as one of the main conclusions). Also, only one pair of temperatures (FT-1, FT-2) is presented, so we don't actually get any information about the effect of temperature on the survival and further growth of these unknown clusters.

The measured GRs at FT-2 are high, 15-23nm/h, so it is very hard for me to believe that they could be caused by SO2 and O3 only, especially as the particles are smaller than 2nm when they exit the nucleation tube. Rather, it seems possible (as the authors also mention) that organic contaminants (which were not measured) could affect the growth either directly and/or by participating in forming sulfuric acid. This would also explain the observed nucleation in the growth tube. The possible effect of bases on growth of sub-3nm particles, found in several recent studies, is not discussed here at all. Therefore, I don't think that there is enough justification to speculate on unknown heterogenous reactions causing the growth, as the authors claim in the abstract and conclusions.

I'm looking forward to the final results of the TANGENT study, as I think there is potential to do more and get interesting results with this setup, but I don't think the initial results as presented here should warrant publication in ACP.

More specific comments: -The authors should give realistic uncertainly estimations for their results and think about the accuracy of the values presented. Especially figure

[Figure]
* * *
Interactive
comment

3 should have error bars, and it should be discussed what kind of error estimates this gives for the critical diameter and growth rates. -The calculation method of nucleation and growth rates should explained better, as well as the meaning of the growth rate factor. The variables used in the equations (and where the values for them come from) should be explained.

---

## Referee Comment (RC2) · Anonymous Referee #2 · 14 Feb 2019

**Review of Tiszenkel et al, Temperature effects on sulfuric acid aerosol nucleation and growth: initial results from the TANGENT study**

**Scientific significance**

New particle formation is a difficult and important problem, and there is currently a lack of diversity in the laboratory studies used to characterize it. Measurements from independent groups are to be encouraged.

However, the current manuscript requires revision if it is to be suitable for ACP. At first glance, it is unclear how much new information it adds to Yu et al (2017), although the new results do become clearer with a very careful reading. The last part of the introduction should be expanded to explain how the new study differs (more measurement data in the J vs temperature parameter space, and the addition of the second flow tube). The paper does not explicitly explain the point of the TANGENT apparatus, nor why it is an improvement on previous experimental setups, except via the sentence in the abstract that it allows nucleation and growth to be studied *independently*. This sentence should be revisited in the text with a better explanation for why this is an improvement on Yu et al (2017) where nucleation and growth rates are presented *separately* already. I appreciate that "separately" and "independently" are not the same, but this needs to be made more obvious in the paper text.

The lack of direct measurements of contaminant NH3 and amines during the experiments is a serious shortcoming, as the concentrations of contaminants could differ markedly between 2017 and 2018 measurement periods. This shortcoming limits the quantitative usefulness of the results, and places high demands on the quality of the data analysis and presentation if the paper is to meet the ACP publication criteria.

In addition to explaining explicitly the benefits of their new setup, the authors should consider setting their paper apart by including in their figures a more detailed, quantitative comparison with other relevant literature, for example Duplissy et al (2016) or Dunne et al (2016). The authors could try to determine from published nucleation measurements what ammonia or amine concentration would be required to reproduce the new particle formation rates they measure.

In order to make clear the usefulness of the TANGENT setup, the authors should explain explicitly and quantitatively how they can put several different concentrations of precursors in flow tube 2, or maintain them at different temperatures, and measure different growth rates, for a constant nucleation rate in flow tube 1. This is exactly what is done in Figure 6 – which is excellent. However, the figure is presented in the text as describing the situation with different temperatures in the two flow tubes. While the temperatures in the flow tubes happened to be different in the measurement presented, the figure actually describes the effect of varying ozone, and no quantitative conclusion about the effect of temperature can be extracted from it. To quantify the effect of temperature in flow tube 2, another figure is needed where the data in Figure 6 are compared to a corresponding measurement in which the two flow tubes are kept at the same temperature.

More generally, the existing plots show the parameter space of nucleation rate vs sulfuric acid, temperature and humidity is quite well explored, but this could have been achieved without the second flow tube and similar measurements were already published by Yu et al (2017). It would be useful to present more measurements where the conditions in the second flow tube are varied with those in the first tube fixed.

**Scientific quality**

The measurements and calculations of nucleation rate use techniques which have been published previously. The experimental apparatus is described clearly. The quality of the data is therefore reasonably well-established, apart from the lack of measurements of contamination that I already mentioned. I have only a couple of outstanding questions.

What are the temperature and RH dependences of the critical cluster size? Why is critical cluster size equal to the diameter at [H2SO4]=0 (please add reference)?

The survival of the particles in the second flow tube is clearly difficult to disentangle from the strange additional growth via sulfur dioxide and ozone. I don't have any good ideas for why this nucleation happens, beyond the obvious speculations about unmeasured contamination. Could the SO2+O3 reaction be because of a contamination by alkaline material – metallic fragments for example, or enormously high amine or ammonia concentrations – which raise the pH to something like what is seen in sea spray aerosols or cloud droplets?

**Presentation quality**

The written English is generally of good quality. There are a few missing articles "a" and "the" distributed through the text.

The sentence "Larger mean diameters were detected under lower temperatures for a given [h2SO4]", would imply the method used to determine the critical cluster size would give a larger critical cluster size for lower temperatures. It is clear from theory and from Figure 3 that this is not the case, so the sentence could be rephrased.

Figures 3 and 4 need error bars, if possible, or at least a careful explanation of what the uncertainties are, what is in the noise and what is a real effect.

"Our results thus show that particles were observed at the end of the room temperature nucleation tube after they were initially nucleated at lower temperatures growth tube. These results can explain the presence of newly formed particles observed in Amazon forests by (Wang et al., 2016),…..
"
It is not clear that the second sentence follows from the first. The focus on the Amazon here and in the introduction and conclusion seems odd, since this is one of the few locations on Earth where nucleation may not be dominated by the sulfuric acid clustering that is the subject of this paper. The demonstration that particles survive when the temperature increases is useful, however, and this enabled me to understand the reason for the TANGENT setup. The same mechanism that operates in the Amazon also operates in marine regions, where sulfuric acid nucleated in the upper troposphere survives to make CCN at cloud level (see recent papers by Lynn Russell's group from the North Atlantic, or much earlier work by Tony Clarke and others).

---

## Author Comment (AC1) · 16 Apr 2019

Thank you very much for your thorough responses and suggestions for this manuscript. Attached to this reply is a zip file containing the specific replies to your comments, a pdf copy of the revised manuscript, and a pdf copy of the tracked changes between the original and revised manuscript.

We hope that our revisions address your comments sufficiently. We appreciate any further questions, comments or concerns that you may have.

Thank you, Lee Tiszenkel

[Figure]

Please also note the supplement to this comment:
https://www.atmos-chem-phys-discuss.net/acp-2019-3/acp-2019-3-AC1-supplement.zip

---

## Author Response (AR1)

**Response to RC1: 'Review of Tiszenkel et al.', Anonymous Referee #1, 03 Feb 2019**

The article is nicely written, relevant literature is cited and the shortcomings of the current study (mainly related to the experimental setup) are discussed appropriately. However, I don't think the study brings anything new to the current scientific literature. The temperature effects on nucleation and early growth have already been discussed extensively in the previous

5    study from partly the same author group (Yu et al. 2017), as well as in the studies from the CLOUD community.

*Response 1: We appreciate the reviewer's helpful comments. In our revision, we elaborate on the following three points that stress the new material that this study brings to the scientific literature. By adding these new analysis and discussions, the manuscript has improved significantly.*

10    1) *The TANGENT apparatus: CLOUD (Kurten et al., 2016, Duplissy et al., 2016, Kirkby et al., 2011) and Yu 2017's experiments focus on nucleation and growth of sub-3 nm particles in a consistent environment. In this study, due to the unique experimental setup allowed by TANGENT, we are examining nucleation and further growth independently, which has not yet been published. For example, the experiments in Figures 6 and 7 are unique to this study and require the TANGENT experimental setup to accomplish.*

15    2) *Temperature effects on growth: very few studies exist that elaborate on temperature effects on growth of newly formed particles (Skrabalova et al., 2014, Yu et al., 2017). The TANGENT experimental setup allows for unique experiments where conditions for nucleation can vary in temperature, allowing for study of subsequent growth of particles formed at a variety of temperature conditions. The experiments undertaken in this study illustrate that new and unique experimental design.*

20    3) *Parameter space: This study contributes more measurement data of nucleation and growth rates in a wide parameter space of temperature and RH conditions, compared to Yu et al., 2017.*

4)

Given that the current study lacks crucial information about the cluster composition and measurements of base organic contaminations in their system (thus relying on many assumptions based on comparison to the previous literature), there is

25    little value in the few additional data points and speculations that are presented.

*Response 2: We understand the limitations of the experimental measurements available to us for this study. We have made the revised manuscript more robust based on the three points in response 1 by including a more thorough discussion section with a broader scope of applications of this data (Specifically the applicability of this data to the marine boundary layer and*

30    *polluted megacities, Section 3.2, page 11 and 12), additional literature review (specifically the growth effects of bases present in the system as well as a discussion of cluster composition, section 3.2, page 10 and 11), and additional data (Fig 7).*

Given their high nucleation and growth rates for each sulfuric acid concentrations in the nucleation tube, I agree with the authors, that the nucleation mechanism is most probably ternary, i.e. involving ammonia and/or amines. However, the authors have not measured the cluster composition at the end of the nucleation tube, which, most probably, critically affects the survival

35    of the nucleated clusters in warmer temperatures (what they intended to study and which is presented as one of the main conclusions).

*Response 3: We agree. We have added a more thorough discussion of chemical composition to the discussion section (Section 3.2, page 11) in order to account for the question of chemical composition effecting survivability. By doing so, we found that the possible chemical composition that we estimated based on CLOUD studies are actually consistent with high GR in FT-2, which we also believe in large part are enhanced by base-multicomponent effects proposed by (Lehtipalo et al. 2018).*

Also, only one pair of temperatures (FT-1, FT-2) is presented, so we don't actually get any information about the effect of temperature on the survival and further growth of these unknown clusters.

*Response 4: This is a good point. We have created a new figure (Figure 7 in manuscript) with FT-1 and FT-2 both at the same temperature to serve as a contrast to the temperature gradient seen in Figure 6. This data is intended to address the shortcoming of the previous draft that did not show the effects of nucleation temperature on survival and growth of newly formed particles. By including this new figure, the revised manuscript shows more clearly the temperature effects on growth of the newly formed particles (Section 3.2, page 10).*

The measured GRs at FT-2 are high, 15-23nm/h, so it is very hard for me to believe that they could be caused by SO2 and O3 only, especially as the particles are smaller than 2nm when they exit the nucleation tube. Rather, it seems possible (as the authors also mention) that organic contaminants (which were not measured) could affect the growth either directly and/or by participating in forming sulfuric acid. This would also explain the observed nucleation in the growth tube.

*Response 5: We recognize that these measured GRs are high. To that end, we have deduced three possibilities for these high growth rates: organic contaminants leading to growth by HOMs as parameterized by Trostl 2016, multicomponent growth as parameterized by Lehtipalo 2018, and cluster/cluster collisional growth as shown by Lehtipalo 2016 (Section 3.2, page 10-11).*

The possible effect of bases on growth of sub-3nm particles, found in several recent studies, is not discussed here at all. Therefore, I don't think that there is enough justification to speculate on unknown heterogenous reactions causing the growth, as the authors claim in the abstract and conclusions.

*Response: We included new refs. (Lehtipalo et al., 2018 and 2016; Stolzenburg et al., 2018) that we unfortunately overlooked previously. Our new analysis shows that it is very likely that the growth of the particles in these experiments included base stabilization as an important mechanism (Section 3.2, page 10-11). But we still believe that ozone and $SO_2$ also contribute to the additional nucleation in FT-2, and we added more discussion.*

I'm looking forward to the final results of the TANGENT study, as I think there is potential to do more and get interesting results with this setup, but I don't think the initial results as presented here should warrant publication in ACP.

More specific comments: -The authors should give realistic uncertainly estimations for their results and think about the accuracy of the values presented. Especially figure 3 should have error bars, and it should be discussed what kind of error estimates this gives for the critical diameter and growth rates. –The calculation method of nucleation and growth rates should

explained better, as well as the meaning of the growth rate factor. The variables used in the equations (and where the values for them come from) should be explained.

*Response 7: Thank you – we agree and we have addressed them in the revised manuscript. We have elaborated on the calculation of nucleation and growth rates and have given a more complete definition of growth rate factor, with explicit definitions of all variables in equations (Section 2.2, page 6-7). Uncertainty analysis is also included in a new section (2.3, page 7) now.*

**Response to RC2: 'Review of Tiszenkel et al, Temperature Effects on Sulfuric Acid Aerosol Nucleation and Growth: Initial Results from the TANGENT Study', Anonymous Referee #2, 14 Feb 2019**

**Scientific significance**

New particle formation is a difficult and important problem, and there is currently a lack of diversity in the laboratory studies

5 used to characterize it. Measurements from independent groups are to be encouraged. However, the current manuscript requires revision if it is to be suitable for ACP. At first glance, it is unclear how much new information it adds to Yu et al (2017), although the new results do become clearer with a very careful reading. The last part of the introduction should be expanded to explain how the new study differs.

10 *Response 1: We thank the reviewer's for thoughtful comments. We agree that there was a lack of clarity in the original manuscript of the new elements of this study as it builds upon previous work, notably Yu et al 2017 which examined a similar parameter space in a similar setup. To that end, we have expanded both the introduction (Page 3, lines 8-22) and the discussion section (Section 3.2, final 2 paragraphs) to stress how this study, especially using the new TANGENT experimental setup, can contribute new findings to the literature.*

The paper does not explicitly explain the point of the TANGENT apparatus, nor why it is an improvement on previous experimental setups, except via the sentence in the abstract that it allows nucleation and growth to be studied independently. This sentence should be revisited in the text with a better explanation for why this is an improvement on Yu et al (2017) where nucleation and growth rates are presented separately already. I appreciate that "separately" and "independently" are not the

20 same, but this needs to be made more obvious in the paper text.

*Response 2: We agree. In the revised manuscript, we added an expanded statement on the combination of FT-1 and FT-2 in the experimental setup section to clarify this further (Section 2.1, page 4 and 5). And in the results sections, we also included more data analysis of TANGENT results, and added discussion on the implications on the data. These TANGENT results are*

25 *now presented in a separate section (Section 3.2) from FT-1 only results (now Section 3.1).*

The lack of direct measurements of contaminant NH3 and amines during the experiments is a serious shortcoming, as the concentrations of contaminants could differ markedly between 2017 and 2018 measurement periods. This shortcoming limits the quantitative usefulness of the results, and places high demands on the quality of the data analysis and presentation if the

30 paper is to meet the ACP publication criteria.

*Response 3: We make several assumptions in our data analysis with regard to ammonia and amine contamination.*
*First, by ensuring that our experimental conditions, including using the same flow tubes, adhering to the same cleaning technique/schedule, using the same suppliers for any species added to the system, and running experiments with the same*

*precursors, we can still be confident that the contaminant levels should not have dramatically changed between 2017 and 2018.*

*Second, 2017 ammonia and amine measurements were conducted both in the UD laboratory and in UAH's laboratory before transporting all instruments for the IOP, and the ammonia/amine readings in both environments were similar, showing ammonia and amine levels very close to the detection limit of the instrument (ranging from 1 to 40 pptv depending on the species being measured). This consistency between different environments leads us to believe that ammonia/amine measurements from the 2017 IOP can give us a good idea of measurements from the 2018 IOP experiments.*

In addition to explaining explicitly the benefits of their new setup, the authors should consider setting their paper apart by including in their figures a more detailed, quantitative comparison with other relevant literature, for example Duplissy et al (2016) or Dunne et al (2016). The authors could try to determine from published nucleation measurements what ammonia or amine concentration would be required to reproduce the new particle formation rates they measure.

*Response 4: A quantitative comparison with CLOUD data is an important aspect of this study. We modified Figure 4 with CLOUD data from Dunne et al (2016), using data points of neutral nucleation. We believe it is now clearer where this study results stand in the current knowledge of the field.*

In order to make clear the usefulness of the TANGENT setup, the authors should explain explicitly and quantitatively how they can put several different concentrations of precursors in flow tube 2, or maintain them at different temperatures, and measure different growth rates, for a constant nucleation rate in flow tube 1.

*Response 5: We agree. As stated in response 2, the experimental setup section of the manuscript was expanded to specifically address the questions raised here. Specifically, we have addressed that FT-2 is kept at constant T and RH and only ozone was varied in FT-2. FT-1 was varied more diversely, with temperature, RH and $SO_2$ varied across experiments (Section 3.1). This had the effect of varying $SO_2$ in FT-2 after dilution. These experiments aimed to measure two effects in the system – the effect of changing temperature in the nucleation region as well as the effect of varying ozone in the growth region (Section 3.2).*

This is exactly what is done in Figure 6 – which is excellent. However, the figure is presented in the text as describing the situation with different temperatures in the two flow tubes. While the temperatures in the flow tubes happened to be different in the measurement presented, the figure actually describes the effect of varying ozone, and no quantitative conclusion about the effect of temperature can be extracted from it.

*Response 6: Yes, we address this together with the following comment.*

To quantify the effect of temperature in flow tube 2, another figure is needed where the data in Figure 6 are compared to a corresponding measurement in which the two flow tubes are kept at the same temperature. More generally, the existing plots

show the parameter space of nucleation rate vs sulfuric acid, temperature and humidity is quite well explored, but this could have been achieved without the second flow tube and similar measurements were already published by Yu et al (2017). It would be useful to present more measurements where the conditions in the second flow tube are varied with those in the first tube fixed.

*Response 7: We agree. We have prepared an additional figure, Figure 7 in the manuscript, that shows results from an experiment that was done with FT-1 and FT-2 at the same temperature. The discussion section was expanded to discuss the implications of the comparison between the results with a temperature gradient and the results with a uniform temperature throughout the system (Section 3.2, page 10). In short, by adding this new figure, we can see more clearly that the clusters can*

10   *survive evaporation when they are transferred between different temperature regions.*

**Scientific quality**

The measurements and calculations of nucleation rate use techniques which have been published previously. The experimental apparatus is described clearly. The quality of the data is therefore reasonably well-established, apart from the lack of

15   measurements of contamination that I already mentioned. I have only a couple of outstanding questions. What are the temperature and RH dependences of the critical cluster size?

*Response 8: The temperature dependence on critical cluster size was a linear correlation ($R^2$ = 0.98), going from 1.627 nm at 258 K to 1.651 nm at 297 K. However, considering an error of ±0.2 nm in these measurements, it is indeed difficult to make a*

20   *definitive conclusion. The RH dependence is more difficult to surmise as RH was difficult to control in the nucleation region, but the critical cluster diameter is negatively correlated with RH across the temperature range; for example, at 268 K the critical cluster diameter was calculated at 1.50 nm at 80% RH and 1.69 nm at 23% RH. Again, the amount of error makes this fairly inconclusive.*

25   Why is critical cluster size equal to the diameter at [H2SO4]=0 (please add reference)?

*Response 9: This assumption is based on the equation for growth rate factor used in Yu et al 2017. The equation used to calculate growth rate factor (that is, enhancement of the growth rate over 1 ppt $H_2SO_4$ leading to 1 nm $h^{-1}$ growth), is:*

$$k_G = \frac{\Delta D_{p,tr} \times 10^7 cm^{-3}}{[H_2SO_4]_0} \frac{k_L}{1 - e^{-nk_L t_r}}$$

30   $\Delta D_{p,tr}$ *represents the particle growth after nucleation; therefore when* $\Delta D_{p,tr} = 0$*, no growth has occurred past nucleation, and therefore the* $D_p$ *at that point is the critical radius. In figure 3, the equation of the fit lines for each temperature is* $D_p = \frac{\Delta D_{p,tr}}{\Delta [H_2SO_4]_0} ([H_2SO_4]) + b$*. If* $\Delta D_{p,tr} = 0$ *then* $D_p$ *represents the critical radius, which equals the y-intercept of the line. We have now included this clarification (Section 2.2, Page 6 and 7)*

The survival of the particles in the second flow tube is clearly difficult to disentangle from the strange additional growth via sulfur dioxide and ozone.

*Response 10: Indeed, the growth rates in FT-2 are high. We believe there are some heterogeneous process involving SO2 and ozone are contributing to the additional nucleation and growth in FT-2. However, we do not understand these chemical mechanisms at present. Additionally, regarding to the growth, it seems that base contaminations in FT-2 are partially responsible, based on findings from Lehtipalo et al. (2018 and 2016) studies. We included this new discussion in Section 3.2.*

I don't have any good ideas for why this nucleation happens, beyond the obvious speculations about unmeasured contamination. Could the SO2+O3 reaction be because of a contamination by alkaline material – metallic fragments for example, or enormously high amine or ammonia concentrations – which raise the pH to something like what is seen in sea spray aerosols or cloud droplets?

*Response 11: The presence of transition metals in the experimental setup could indeed serve as a source for oxidation at higher pHs (Seinfeld and Pandis p .294): The funnels at the beginning and end of FT-2 were stainless steel. It is unclear how heterogeneous reactions of our precursor gases on the surface of the flow tube could impact the nucleation. However, the particles in this experiment are likely acidic as can be seen from cluster composition of base nucleation as shown in CLOUD experiments (Kirkby 2011; Almeida 2013). Lawler et al., 2016 also showed acidic chemical composition of nanoparticles with sulfuric acid and base nucleation. We added new discussions in Section 3.2 to address acidity of the particles.*

**Presentation quality**

The written English is generally of good quality. There are a few missing articles "a" and "the" distributed through the text. The sentence "Larger mean diameters were detected under lower temperatures for a given [h2SO4]", would imply the method used to determine the critical cluster size would give a larger critical cluster size for lower temperatures. It is clear from theory and from Figure 3 that this is not the case, so the sentence could be rephrased.

*Response 12: We have fixed these errors.*

Figures 3 and 4 need error bars, if possible, or at least a careful explanation of what the uncertainties are, what is in the noise and what is a real effect.

*Response 13: We agree. And we added new Section 2.3 and discussed detailed error propagation analysis.*

"Our results thus show that particles were observed at the end of the room temperature nucleation tube after they were initially nucleated at lower temperatures growth tube. These results can explain the presence of newly formed particles observed in

Amazon forests by (Wang et al., 2016),….. " It is not clear that the second sentence follows from the first. The focus on the Amazon here and in the introduction and conclusion seems odd, since this is one of the few locations on Earth where nucleation may not be dominated by the sulfuric acid clustering that is the subject of this paper.

5 *Response 14: We agree that those two sentences in the original manuscript did not come together as clearly as we had intended, and that focusing on the Amazon is shortsighted considering our results. We have added some clarifying contents between the two sentences in the revised manuscript. We have addressed the focus on the Amazon boundary layer by broadening the scope of our study, discussing our results in the context of the marine boundary layer, where $H_2SO_4$ particle formation certainly occurs, as well as polluted megacities, where NPF occurs despite high pre-existing particle loads with abundant pollutant*
10 *species present such as $SO_2$ and $O_3$, as the conditions in FT-2 were during these experiments (Section 3.2).*
*Regardless of the composition of the particles, studies of how particles evolve once they are transferred between environmental conditions represent an area of aerosol nucleation and growth that lacks laboratory study, and this manuscript represents experiments and observations that can initiate further investigation in this area.*

15 The demonstration that particles survive when the temperature increases is useful, however, and this enabled me to understand the reason for the TANGENT setup. The same mechanism that operates in the Amazon also operates in marine regions, where sulfuric acid nucleated in the upper troposphere survives to make CCN at cloud level (see recent papers by Lynn Russell's group from the North Atlantic, or much earlier work by Tony Clarke and others).

20 *Response 15: We appreciate this comment. We have added a more thorough discussion of the implications of the results from TANGENT to Section 3.2, addressing marine boundary layer aerosol distributions as well as a more detailed description of how these results can help to explain new particle formation in areas such as polluted megacities where NPF occurs despite a high preexisting particle load.*

[revised manuscript text omitted]
$_2$SO$_4$]$_0$ at the temperature between 258 and 297 K. [H$_2$SO$_4$]$_0$ was varied from 8 ̶x̶ ×$10^6$ cm$^{-3}$ to 7 ̶x̶ ×$10^7$ cm$^{-3}$. The RH was kept in relatively narrow range between 20% and 30%. The y-intercept in Fig. 3 indicates a̶the critical cluster diameter o̶f̶ was estimated to be between 1.6̶7̶6̶-1.6̶8̶ ̶n̶m̶.7 nm depending on temperature, with lower temperatures resulting in smaller critical cluster diameters. However,

15   since the PSM inversion that determines $D_p$ has an uncertainty of ±0.2 nm, it is difficult to discuss the temperature trends of the critical cluster diameter. This critical size is consistent with K̶u̶l̶m̶a̶l̶a̶ ̶e̶t̶ ̶a̶l̶.̶ ̶(̶2̶0̶1̶3̶)̶(Kulmala et al., 2013) and A̶l̶m̶e̶i̶d̶a̶ ̶e̶t̶ ̶a̶l̶.̶ ̶(̶2̶0̶1̶3̶)̶,̶(Almeida et al., 2013), which determined critical cluster diameters of 1.5 + 0.3 nm and 1.7 nm, respectively. L̶a̶r̶g̶e̶r̶ ̶m̶e̶a̶n̶ ̶d̶i̶a̶m̶e̶t̶e̶r̶s̶ ̶w̶e̶r̶e̶ ̶d̶e̶t̶e̶c̶t̶e̶d̶ ̶u̶n̶d̶e̶r̶ ̶l̶o̶w̶e̶r̶ ̶t̶e̶m̶p̶e̶r̶a̶t̶u̶r̶e̶s̶ ̶f̶o̶r̶ For a given [H$_2$SO$_4$]$_0$.̶, the mean $D_p$ at the end of the 45 second residence time in FT-1 was larger for lower temperatures. Previously, (Glasoe et al., 2015;Yu et al., 2017a) have also shown

20   increasing $GR$ with increasing [H$_2$SO$_4$] from flow tube experiments. The slope of $D_p$ vs. [H$_2$SO$_4$]$_0$ increased with each 10 degree decrease in temperature over the course of these experiments. Thus, the growth rate factor $k_G$ also increased with subsequent temperature decreases (e.g., from 1.27 at 297 K to 12.6 at 258 K). These results indicate that lower temperatures promote the faster growth of particles due to the reduction in saturation vapor pressures of H$_2$SO$_4$ at lower temperatures.

Figure 4 shows the relationship of Log $J$ vs. Log RA for different temperatures. Experiments were conducted at 10 K intervals,

25   starting from 297 K down to 258 K for RH between 41% and 45%. Across all temperature and RH experiments conducted, in general, $J$ values were shifted 2 to 3 orders of magnitude above previous literature values [̶Y̶u̶ ̶2̶0̶1̶7̶,̶ ̶B̶r̶u̶s̶ ̶2̶0̶1̶1̶]̶.of flow tube nucleation studies (Brus et al., 2010;Yu et al., 2017a) and 4 to 5 orders of magnitude above CLOUD measurements of H$_2$SO$_4$ nucleation as shown in the figure using CLOUD data from (Dunne et al., 2016). This upward shift was consistent across trials. Based on our measured NH$_3$ and amine concentrations (Fig. 2), this upward shift is consistent with the nucleation rate

30   enhancement due to synergistic effects of NH$_3$ concentrations on the order of 20 to 30 pptv,̶ and dimethylamine concentrations on the order of 1̶ ̶t̶o̶ 25 ppt reported by other studies (̶G̶l̶a̶s̶o̶e̶ ̶e̶t̶ ̶a̶l̶.̶,̶ ̶2̶0̶1̶5̶)̶. 
[revised manuscript text omitted]

---

## Author Response (AR2)

Response #1

The novelty and possibilities of the TANGENT setup are described better in the revised version, as well as the methods used in the study. More discussion is provided about possible cluster composition and possible growth mechanisms, but I'm still puzzled by the additional nucleation and fast growth in FT2 and how this can be disentangled from cluster survival.

Revision suggestions:

1) Exact NH3 and amine values should be removed from Table 1 and rather just give the general levels estimated based on the earlier study in the text (in my experience it is difficult to control and measure these levels even in the same setup).

*Response: We agree that since these values are estimated from a previous identical setup they do not belong in the table of results from this study, especially considering that they are presented in figure 2. The $NH_3$ and amine values are now discussed in section 3.1 instead of in the table.*

2) The main numerical results from Fig 6 and 7 (growth and survival at different O3 and T difference) should be combined into one figure (in addition to figures 6 and 7).

*Response: We have added a new figure (Fig 8) that displays these values numerically, and we have added it to the discussion of these experiments in section 3.2.*

3) I appreciate the new discussion in section 3.2., but it is rather lengthy and could be shortened and structured better, so that it is more clear what the current results actually tell and what is pure speculation.

*Response: We agree that the speculation at the end of 3.2 is inappropriate for that section, especially considering that it is stated in section 4 along with the implications of the results. We believe that this better distinguishes experimental results from speculation.*

Response #2

5   While the authors were unable to address many of the previous serious criticisms of the paper during its first reviews, the analysis and presentation is significantly improved. The authors took the comments to heart and worked hard to address those that they were able to deal with in the absence of additional experiments.

Minor comments

In the abstract, the authors say "sulfuric acid nucleation….becomes barrierless below 268K". This appears to be conflicting with previous findings, for example, those from the CLOUD experiment by Kuerten et al referred to on page 2 line 21, that nucleation rates of sulfuric acid with very low base contamination can be enhanced by ammonia at much lower temperatures. Perhaps it would be better to say "indicating that particle formation in the conditions of our flow tube takes place via barrierless

15  nucleation at lower temperatures"
Also suggest "in the presence of SO2 and ozone, and potentially other contaminant vapors".
Page 2 line 21: 5105 ->5 <times> 105; line 31 suggest "this"->"their" to distinguish the current and cited studies.
*Response: We agree with these clarifications, and have implemented these suggestions.*
Page 3 line 20: in my previous review I was suggesting the authors refer to much more recent work from the North Atlantic

20  NAAMES campaign, but these citations are still also valuable.
Page 5 i.d.->inner diameter
Page 7 "an identical condition"->"identical conditions"
*Response: We have corrected these errors.*
Page 8 "consistent with"->"consistent with the studies of"; the sentence on lines 29-30 could be better phrased or deleted.

25  *Respons: We have implemented this change which clarifies the statements better. We have removed the sentence on lines 29-30 because it was redundant considering the previous statements.*
Page 9: Duplissy et al say nucleation is "close to kinetic" at 208K –I would not say this was "consistent" with the findings of the present study at 268K. I don't think this matters for the science done in this paper, the sentence on lines 3-4 is simply incorrect and should be deleted. If Duplissy et al had made this assertion, wouldn't it be inconsistent with the findings of

30  Kuerten et al that ammonia strongly enhances particle formation even at low temperatures?
*Response: We have removed this statement as it indeed did not agree with the cited studies.*
Page 9 line 27 remove extraneous "the".
Line 29: is it correct to use "ph" for "sulphate" and "f" in "sulfuric acid"?
*Response: Changed to "sulfate" to be more consistent throughout.*

Page 10 line 4 remove extraneous "a"

*Response: We have corrected this error.*

The last paragraph and the earlier discussion of NPF in megacities seems rather speculative given the lack of understanding of the origin of the apparent SO2+ozone effect. Page 12 line 31 suggest "proposed here, if they can be repeated and fully understood in future studies, may provide some insights…."

*Response: This is an important caveat to make towards our speculative statement about NPF in megacities. We have added it in per your suggestion.*

Figure 2 caption: "as an example one day of measurements"

Figure 7 caption: "a constant"->"the same"

*Response: Thank you for your very careful reading. We have corrected these minor errors.*

[revised manuscript text omitted]